# Metformin Inhibits Lipid Droplets Fusion and Growth via Reduction in Cidec and Its Regulatory Factors in Rat Adipose-Derived Stem Cells

**DOI:** 10.3390/ijms23115986

**Published:** 2022-05-26

**Authors:** Lijing Yang, Xiaowei Jia, Dongliang Fang, Yuan Cheng, Zhaoyi Zhai, Wenyang Deng, Baopu Du, Tao Lu, Lulu Wang, Chun Yang, Yan Gao

**Affiliations:** 1Department of Human Anatomy, School of Basic Medical Sciences, Capital Medical University, Beijing 100069, China; yanglijing2022@126.com (L.Y.); jxw@ccmu.edu.cn (X.J.); sdwffdl@163.com (D.F.); zhaizy0826@163.com (Z.Z.); dengwenyang496@gmail.com (W.D.); dubaopu1989@163.com (B.D.); lu1974tao@sina.com (T.L.); lwang@ccmu.edu.cn (L.W.); 2Beijing Key Laboratory of Cancer Invasion and Metastasis Research, Capital Medical University, Beijing 100069, China; 3Department of Physiology and Pathophysiology, School of Basic Medical Sciences, Capital Medical University, Beijing 100069, China; chengy0312@163.com; 4Department of Experimental Center for Basic Medical Teaching, School of Basic Medical Sciences, Capital Medical University, Beijing 100069, China

**Keywords:** metformin, lipid droplets, adipogenesis, cidec, perilipin1

## Abstract

Metformin is still being investigated due to its potential use as a therapeutic agent for managing overweight or obesity. However, the underlying mechanisms are not fully understood. Inhibiting the adipogenesis of adipocyte precursors may be a new therapeutic opportunity for obesity treatments. It is still not fully elucidated whether adipogenesis is also involved in the weight loss mechanisms by metformin. We therefore used adipose-derived stem cells (ADSCs) from inguinal and epididymal fat pads to investigate the effects and mechanisms of metformin on adipogenesis in vitro. Our results demonstrate the similar effect of metformin inhibition on lipid accumulation, lipid droplets fusion, and growth in adipose-derived stem cells from epididymal fat pads (Epi-ADSCs) and adipose-derived stem cells from inguinal fat pads (Ing-ADSCs) cultures. We identified that cell death-inducing DFFA-like effector c (Cidec), Perilipin1, and ras-related protein 8a (Rab8a) expression increased ADSCs differentiation. In addition, we found that metformin inhibits lipid droplets fusion and growth by decreasing the expression of Cidec, Perilipin1, and Rab8a. Activation of AMPK pathway signaling in part involves metformin inhibition on Cidec, Perilipin1, and Rab8a expression. Collectively, our study reveals that metformin inhibits lipid storage, fusion, and growth of lipid droplets via reduction in Cidec and its regulatory factors in ADSCs cultures. Our study supports the development of clinical trials on metformin-based therapy for patients with overweight and obesity.

## 1. Introduction

As the most common chronic metabolic disease worldwide, obesity develops from a long-term energy imbalance when dietary energy intake exceeds energy expenditure [1,2]. It is a significant risk factor for various diseases, including type II diabetes [3], cardiovascular diseases [3,4], fatty liver [1,2], and several types of cancers [5,6]. The increase in the prevalence of obesity generates an important public health issue. Thus far, the current prevention and treatments have not been successful for long-term treatment [2,7]. The guidelines on obesity recommend that lifestyle interventions, including dietary change and increased physical activity, are the first-line treatment for obesity [8,9,10]. The limitations of lifestyle interventions are poor persistence and weight regain. Pharmacotherapy is recognized as the second-line treatment for patients with obesity without achieving adequate weight loss after lifestyle interventions [8,10]. Although several weight loss drugs (such as semaglutide or orlistat) are approved as monotherapy for obesity treatments [8,11,12], there is a constant need to develop new therapeutic strategies for obesity treatments.

Currently, metformin is recommended as the preferred initial drug for type II diabetes treatments due to its efficacy, safety profile, suitable tolerability, and low hypoglycemic risk [13,14]. There is strong evidence to demonstrate that metformin has favorable effects beyond its action on glycemic control. It has been reported that the weight loss effects of metformin in adults with overweight or obesity and adolescents without diabetes [15]. A large and comprehensive study has shown that weight loss benefits are related to adherence to metformin and persist for up to 10 years [16]. Recent research showed that metformin treatment inhibited body weight and fat mass gain in obese mice on a high-fat diet [17,18]. The American Association of Clinical Endocrinologists/American College of Endocrinology (AACE/ACE) Obesity Clinical Practice Guidelines (2016) (2016) recommend using of metformin in individuals with obesity who have a high risk of type II diabetes or insulin intolerance and failed to respond to lifestyle interventions or anti-obesity pharmacotherapy [19]. Although China Food and Drug Administration (CFDA) and Food and Drug Administration (FDA) have not approved metformin as a weight loss drug, metformin is still being investigated as a potential agent for managing overweight or obesity. Multiple studies have demonstrated possible weight loss mechanisms following metformin treatment in the hypothalamus, adipose-brain axis, gut flora, adipose tissue, and liver [14,20,21]. Recent studies have evidenced that metformin increases circulating levels of growth/differentiation factor 15, which are important for suppressing appetite and promoting weight loss [22,23,24]. A more detailed understanding of metformin will probably lead to advances in optimal treatment for patients with obesity without metabolic complications.

Obesity is characterized by white adipose tissue (WAT) expansion resulting from an excessive accumulation of triglycerides in white adipocytes. WAT expansion involves adipocyte hypertrophy (an increase in adipocyte size) and hyperplasia (an increase in adipocyte number). Adipocyte hyperplasia (adipogenesis) is a process of adipocyte precursors proliferation and differentiation into mature adipocytes [25,26,27]. In addition to hypertrophy, it has been demonstrated that adipogenesis plays an important role in the progression of obesity [28,29,30,31]. Spalding and colleagues have reported that the total number of new adipocytes added per year is significantly greater in individuals with obesity compared with lean individuals [32]. In rodents, high-fat diet feeding induces adipocyte precursors adipogenesis in perigonadal white adipose tissue [31,33]. This evidence suggests a crucial role for adipocyte precursors adipogenesis in human obesity. Therefore, inhibiting the adipogenesis of adipocytes precursors may be a new therapeutic target for obesity treatments. Whether adipogenesis is also involved in the weight loss mechanisms by metformin is not fully elucidated. 

Given that the inguinal and epididymal fat depots were representatives of subcutaneous (SAT) and visceral white adipose tissues (VAT) individually, we, therefore, used adipose-derived stem cells (ADSCs) from inguinal and epididymal fat pads to investigate the beneficial effects and molecular mechanisms of metformin in vitro. In the present study, we investigated lipid accumulation, lipid droplets fusion, and growth in ADSCs exposed to metformin. We also observed the expression of cell death-inducing DFFA-like effector c (Cidec), Perilipin1, and Rab8a, the linchpin and important regulators for lipid formation, in ADSCs cultures after metformin treatments. In addition, we examined the underlying mechanisms of metformin on lipid droplets fusion and growth in ADSCs cultures. Our current study aims to determine the beneficial effects and molecular mechanisms of metformin on adipogenesis in ADSCs cultures.

## 2. Results

### 2.1. Metformin Decreases Lipid Accumulation in Both Ing-ADSCs and Epi-ADSCs Cultures

To investigate the effects of metformin on ADSCs adipogenesis, we firstly performed a measurement of lipid accumulation of ADSCs cultures incubated with serial doses of metformin. As shown in Figure 1A, the majority of adipose-derived stem cells from epididymal fat pads (Epi-ADSCs) were able to differentiate into mature adipocytes under induction conditions. Lots of smaller lipid droplets were present in Epi-ADSCs cultures with metformin treatments at high concentrations, including 2.0 mM, 4.0 mM, and 8.0 mM. This observation indicates that treatment with metformin can result in a dose-dependent decrease in lipid accumulation in Epi-ADSCs cultures. This dose-dependent manner was consistent with the size and distribution of lipid droplets. The percentage of larger lipid droplets (>3 µm) was significantly decreased in Epi -ADSCs cultures exposed to metformin at serial concentrations, including 0.5 mM (46.8%), 1.0 mM (34%), 2.0 mM (23.1%), 4.0 mM (1.7%), and 8.0 mM (0%), when compared with control group (65.4%). (Figure 1B). Accordingly, a significantly increased proportion of smaller lipid droplets (<1 µm) was detected among groups (Figure 1B; control: 0%; 0.5 mM: 21.2%; 1.0 mM: 28.7%; 2.0 mM: 27.3%; 4.0 mM: 32.6%; 8.0 mM: 46.6%). In addition, the lipid droplets diameter was significantly decreased in Epi-ADSCs cultures exposed to metformin at serial concentrations when compared with control group (Figure 1C; 0.5 mM: 3.52 ± 0.13 µm; 1.0 mM: 2.71 ± 0.11 µm; 2.0 mM: 2.12 ± 0.06 µm; 4.0 mM: 1.52 ± 0.05 µm; 8.0 mM: 1.24 ± 0.04 µm; control: 5.38 ± 0.19 µm). These observations were further supported by the quantification of Oil Red O staining and triglyceride in vitro. As shown in Figure 1D, metformin treatments significantly reduced the optical density of Oil Red O staining dissolved in isopropanol. Moreover, the data for triglyceride assay showed that triglyceride deposition was significantly reduced from 0.738 ± 0.052 µmol/mg (control) to 0.507 ± 0.03 µmol/mg (1 mM), 0.275 ± 0.009 µmol/mg (2 mM) and 0.178 ± 0.007 µmol/mg (4 mM) after metformin treatments (Figure 1E). The effects of metformin in Ing-ADSCs cultures were also examined (Figure 1F–J). Metformin treatments resulted in similar phenotypes of lipid accumulation in adipose-derived stem cells from inguinal fat pads (Ing-ADSCs) cultures, including the increased percentage of smaller lipid droplets (Figure 1F,G), decreased levels of lipid droplets diameter (Figure 1H) and triglyceride deposition (Figure 1I,J). 

### 2.2. Metformin Decreases Adipogenic Genes and Lipogenic Enzymes Expression in ADSCs Differentiation

Next, we evaluated the effect of metformin on the cell viability of ADSCs with a cell counting kit 8 (CCK8) assay. The data for cell viability showed that no significant difference was detected in Epi-ADSCs and Ing-ADSCs cultures with metformin at the concentrations of 1 mM, 2 mM, and 4 mM when compared with the control group, respectively (Figure 2A,D). However, 8 mM metformin administration resulted in a significantly decreased level of cell viability in Ing-ADSCs and Epi-ADSCs cultures when compared with the control group (Figure 2A,D). Based on the data of lipid accumulation and cell viability, 2 mM and 4 mM metformin were then used preliminarily to examine its effects on ADSCs cultures. 

We further assess the expression of adipogenic transcription factors in ADSCs cultures after metformin treatments. The mRNA levels of peroxisome proliferator-activated receptor γ (Pparγ) and CCAAT enhancer-binding protein α (Cebpα), key adipogenic transcription factors of adipocyte differentiation, presented a significant decrease in Epi-ADSCs with metformin treatments when compared with the control group (Figure 2B). Significantly reduced levels of mRNA expression of Adipoq (adiponectin gene) and Fabp4 (fatty acid-binding protein 4) were also detected in Epi-ADSCs cultures with 2 mM and 4 mM metformin treatments relative to those in the control group (Figure 2B). Additionally, the mRNA levels of lipogenic enzymes in Epi-ADSCs cultures, including acetyl-CoA carboxylase (Acc), fatty acid synthesis (Fasn), and diacylglycerol acyltransferase 2 (Dgat2), were significantly lower in the metformin group (4 mM) when compared with the control group, respectively (Figure 2C). Similarly, the mRNA levels of key adipogenic transcription factors (Pparγ and Cebpα), adipokines (Fabp4 and Adipoq), and lipogenic enzymes (Acc, Fasn, and Dgat2) showed significantly reduced levels in Ing-ADSCs cultures after metformin treatments (Figure 2E,F). 

### 2.3. Metformin Inhibits Lipid Droplets Fusion and Growth in ADSCs Differentiation

Owing to higher sensitivity and resolution than Oil Red O, Bodipy was then used to detect small lipid droplets to confirm the effects of metformin on lipid droplets fusion and growth. Numerous smaller lipid droplets were observed in Epi-ADSCs and Ing-ADSCs cultures with metformin treatments at the concentrations of 2 mM and 4 mM (Figure 3A,E, upper panel). This phenotype was further confirmed by the immunostaining of Perilipin1 (Figure 3A,E, lower panel). A one-factor ANOVA showed that significantly reduced levels of lipid droplets diameter per cell were present in Epi-ADSCs culture exposed to metformin at 2 mM (1.167 ± 0.042 µm) and 4 mM (0.878 ± 0.027 µm) when compared with the control group (5.748 ± 0.88 µm), respectively (Figure 3B). Therefore, the number of lipid droplets per cell and percentage of smaller lipid droplets (<1 µm) were significantly increased in Epi-ADSCs culture exposed to metformin at 2 mM and 4 mM, when compared with the control group, respectively (Figure 3C,D). Similar tendencies were observed in Ing-ADSCs cultures after metformin treatments, including the lipid droplet diameter per cell, number of lipid droplets per cell, and percentage of lipid droplets (Figure 3F–H). These data indicate that metformin could inhibit the fusion and growth of lipid droplets in the differentiation process of ADSCs.

### 2.4. Cidec, Perilipin1 and Rab8a Expression Are Decreased in ADSCs after Metformin Treatments

Based on the above data of lipid accumulation, diameter, and number of lipid droplets, we then focus on the possible underlying mechanism of inhibition effect of metformin on lipid droplets fusion and growth. Li Peng’s group has reported that lipid droplets fusion and growth are mediated by the cell death-inducing DFFA-like effector (CIDE) family proteins and two regulators [34,35,36,37,38,39,40]. Therefore, we firstly investigate the gene expression of Cidec, Perilipin1, and ras-related protein 8a (Rab8a) in the differentiation process of ADSCs. In differentiated Epi-ADSCs, the mRNA expression of Cidec and Periipin1 precedes its protein expression. The protein expression of Cidec was rapidly increased on day 4 and then maintained a higher level on days 6–8. The protein levels of Periipin1 were detected on days 4–8. The mRNA and protein levels of Rab8a were significantly increased in the differentiation process of Epi-ADSCs. The higher levels of Pparγ mRNA and protein were detected on days 2–8 and days 4–8 in differentiated Epi-ADSCs, respectively (Figure 4A,B). In differentiated Ing-ADSCs, the mRNA and protein expression of Cidec were rapidly increased at the earlier stage (days 2–4) and then maintained a higher level at the later stage (days 6–8). The protein levels of Perilipin1 and Rab8a were significantly increased on days 4–8. The mRNA levels of Perilipin1 and Rab8a were significantly increased on days 2–8. From day 2 to day 8, the expression of Pparγ mRNA and protein was detected and increased gradually in differentiated Ing-ADSCs (Figure 4E,F). 

The data showed that the protein expression of Cidec, Perilipin1, and Rab8a was significantly reduced in differentiated Epi-ADSCs and Ing-ADSCs by 4 mM metformin treatments. Pparγ protein expression was also significantly decreased in differentiated Epi-ADSCs by 2 mM and 4 mM metformin treatments and in differentiated Ing-ADSCs by 4 mM metformin treatment (Figure 4C,G). Similar patterns were observed in mRNA expression of Cidec, Perilipin1, Rab8a, and Pparγ in differentiated Epi-ADSCs and Ing-ADSCs by 4 mM metformin treatments (Figure 2B,E and Figure 4D,H). Because of decreased protein and mRNA levels of Cidec, Perilipin1, Rab8a, and Pparγ in both Epi-ADSCs and Ing-ADSCs cultures after 4 mM metformin treatments, 4 mM metformin was selected for the following experiments.

### 2.5. Metformin Inhibits Cidec, Perilipin1, and Rab8a Expression Partly via Activation of AMPK Signaling in ADSCs

To investigate the mechanisms underlying inhibition of metformin on Cidec and its regulatory factors expression, we evaluated AMPK signaling in ADSCs. As shown in Figure 5A, the levels of AMPKα phosphorylation (p-AMPKα) presented a time course pattern in Epi-ADSCs with 4 mM metformin treatments. DMI alone resulted in a significant increase in the levels of p-AMPKα at 120 min. Additionally, p-AMPKα was further significantly increased by metformin treatments when compared with DMI alone (Figure 5A). A rescue experiment was performed to confirm the AMPK phosphorylation in the inhibition effect of metformin on Cidec, Perilipin1, and Rab8a expression. Compound C (a potent and selective AMPK inhibitor, CC) at 10 µM is used to inhibit AMPK phosphorylation induced by metformin [41]. The protein and mRNA levels of Cidec, Perilipin1, Rab8a, and Pparγ were increased in the metformin + CC group when compared with the metformin group (Figure 5B,C). The morphology of lipid droplets (Figure 5D), the relative optical density (Figure 5E), and average lipid droplet area per cell (Figure 5F) were significantly increased in the metformin + CC group relative to those in the metformin group. Consistent with these results, the mRNA expression of Acc, Fasn, and Dgat2 was also enhanced in the metformin + CC group when compared with the metformin group (Figure 5G). These data suggest that inhibition of AMPK signaling at least in part attenuates the decreased protein and mRNA levels of Cidec, Perilipin1, Rab8a, and Pparγ induced by metformin (Figure 5B–G). In addition, inhibition of AMPK signaling can also partially reverse the decreased levels of lipid droplets fusion, growth, and lipid storage induced by metformin. In differentiated Ing-ADSCs, metformin treatments can also activate the AMPK signaling and inhibit Cidec, Perilipin1, and Rab8a gene expression. Compound C can rescue metformin inhibition on Cidec and its regulatory factors expression. Consistently, Compound C also attenuates metformin inhibition on lipid droplets fusion and growth, and lipogenic enzyme expression (Figure 6A–G). These data indicate that activation of AMPK is involved in metformin inhibition on the fusion and growth of lipid droplets in the differentiation process of ADSCs.

## 3. Discussion

In this study, our results demonstrate the similar effect of metformin in the inhibition of lipid droplets fusion, growth, and lipid storage between Epi-ADSCs and Ing-ADSCs. We identified that Cidec, Perilipin1, and Rab8a expression increased in ADSCs differentiation. Our data show that metformin inhibits lipid droplets fusion and growth by decreasing the expression of Cidec, Perilipin1, and Rab8a. Additionally, activation of AMPK pathway signaling in part involves metformin inhibition on Cidec, Perilipin1, and Rab8a expression. Collectively, our study reveals that metformin inhibits lipid storage, fusion, and growth of lipid droplets via reduction in Cidec and its regulatory factors in ADSCs cultures.

Although it has been acknowledged that inguinal and epididymal fat depots show a region-specific manner, the present study shows a similar tendency of metformin inhibition between Ing-ADSCs and Epi-ADSCs cultures. SAT (including inguinal WAT) is considered relatively beneficial in obesity, whereas VAT (including epididymal WAT) is potentially harmful in obesity and other metabolic disorders [42,43,44]. In addition, adipocytes in inguinal and epididymal fat depots also arise from the distinctive originations during ontogenetic and postnatal development [45,46]. Upon high-fat diet feeding, epididymal WAT expands through both adipocyte hyperplasia and hypertrophy, whereas inguinal WAT expands predominantly through hypertrophy and undergoes a low rate of adipocyte proliferation [31]. By contrast, both Ing-ADSCs and Epi-ADSCs cultures proliferate and robustly differentiate into adipocytes in vitro [47,48,49,50,51]. Currently, the inhibition tendencies of metformin are similar in both Ing-ADSCs and Epi-ADSCs, including the distribution of lipid droplet size, triglyceride deposition, levels of adipogenic transcription factors (Pparγ and Cebpα), and adipokines (Fabp4 and adiponectin). Our data show that metformin can inhibit differentiation and lipogenesis of Epi-ADSCs and Ing-ADSCs in vitro. These findings are consistent with previous studies that metformin inhibits differentiation and lipogenesis of 3T3L1 or human preadipocytes [52,53,54]. However, a dual effect of metformin was recently reported that a lower concentration of (1.25 mM and 2.5 mM) metformin-induced adipogenesis and a higher concentration of metformin (5 mM and 10 mM) significantly inhibited adipogenesis in 3T3-L1 cells [41], which is not consistent with our dose-dependent manner of metformin. Additionally, metformin (500 µm) suppresses adipogenesis in C3H10T1/2 cells [55]. This discrepancy may be due to the different cell models used (cell lines versus primary ADSCs). Metformin inhibition of lipid accumulation is further confirmed by the evidence of increased numbers and decreased area of lipid droplets per cell. The lipid storage capacity of lipid droplets is dependent on their fusion and growth. Therefore, we propose that metformin inhibits lipid droplets fusion and growth in ADSCs differentiation. It is notable that the concentrations of metformin (2 mM and 4 mM) used currently exceed the plasma metformin concentration in humans after oral administration. Therefore, further investigation is required to determine metformin inhibition on lipid droplets fusion in vivo.

Lipid droplets are cytosolic neutral lipid storage depots and function as the central organelle in lipid homeostasis. Lipid droplets fusion and growth occur closely to the differentiation of ADSCs. It has been demonstrated that CIDE family proteins control lipid homeostasis and strongly correlate with obesity [34,37,56,57,58,59,60,61,62,63]. Therefore, we hypothesize that metformin treatments probably lead to decreased expression of CIDE family proteins. Based on the tissue-specific expression of the three CIDE members, we focus on Cidec because of its predominant expression in the WAT [60,64,65,66]. Currently, we found that Cidec expression increased greatly in the process of Epi-ADSCs and Ing-ADSCs differentiation, indicating its close relation with lipid droplets fusion and growth. We further propose that the reduced expression of Cidec contributes to metformin inhibition on lipid droplets fusion and growth, as evidenced by the presence of multiple small lipid droplets and the downregulation of Cidec induced by metformin. Consistently, multiple small lipid droplets were found in WAT from Cidec-KO mice with a regular diet or high-fat diet and a patient with Cidec mutation [64,67], also in differentiated stromal vascular cells from WAT of Cidec-KO mice [36]. These findings strongly support the proposition that metformin can lead to decreased expression of Cidec followed by the formation of multiple small lipid droplets. In addition, several lines of evidence have demonstrated that Perilipin1 and Rab8a are key regulators of Cidec-mediated lipid droplets fusion and growth [34,37,38,57,68]. We, therefore, investigated whether Perilipin1 and Rab8a involve a cooperative action of Cidec during the formation of multiple small lipid droplets induced by metformin. Our data indicate that metformin markedly inhibits the expression of Perilipin1 and Rab8a in Epi-ADSCs and Ing-ADSCs differentiation. The present finding supports previous reports showing that lipid droplet sizes are significantly reduced in Perilipin1-knockdown adipocytes [38], and Rab8a is active in promoting LD fusion and stabilizing the Cidec complex at the lipid droplet contact sites [34,68]. We, therefore, could propose that both Perilipin1 and Rab8a possibly contribute to metformin inhibition on lipid droplets fusion and growth in ADSCs. 

It has been documented that metformin activates AMP-activated protein kinase (AMPK) in in vivo and in vitro experiments [41,55,69,70,71,72]. We next aimed to determine whether AMPK activation is involved in the inhibition effect of metformin on Cidec, Perilipin1, and Rab8a expression. Currently, metformin treatments enhanced AMPK phosphorylation expression and simultaneously inhibited the proteins and mRNA levels of Cidec Perilipin1 and Rab8a expression. Following this, Compound C (AMPK inhibitor) pretreatments provided a partial rescue of decreased Cidec expression and almost complete rescue of decreased Perilipin1 and Rab8a expression induced by metformin. Consistently, Compound C also partly rescued the lipid accumulation, fusion, and growth of lipid droplets, as evidenced by the increased optical density levels of quantified Oil Red O staining and larger areas of lipid droplets. These results were further strengthened by the metformin-induced inhibition of lipogenic enzymes, including Acc, Fasn, and Dgat2, which were also partly reversed by Compound C treatments. Additionally, several groups have reported Pparγ-FSP27/Cidec axis in adipocytes [73,74,75,76]. Inconsistent with these descriptions, we observed that Pparγ expression was downregulated by metformin and rescued by Compound C with a similar pattern as Cidec. This is due to the fact that Cidec expression is transcriptionally upregulated by Pparγ [77]. Our current study provides several lines of evidence that metformin inhibits the expression of Cidec and Perilipin1, correlating to the AMPK pathway in adipose-derived stem cells. It should be noted that blockade of AMPK signaling could not entirely rescue metformin-inhibited expression of Cidec. These data suggest that metformin might decrease Cidec expression via other pathways, such as inflammation and autophagy [78,79,80], integrin/ERK pathway [81], AKT, and MAPK pathways [41,55] The more details of metformin inhibition on Cidec expression still need to be further explored.

Several studies have demonstrated that activation of AMPK can result in the formation of smaller lipid droplets in adipocytes [82,83,84,85,86,87,88]. An AMPK activator (5-Aminoimidazole-4-carboxyamide ribonucleoside, AICAR) leads to smaller lipid droplets appearing in white adipocytes; smaller lipid droplets are one of the morphological features of beige adipocytes [88]. Similarly, activation of AMPK by some natural components or derivations of herbal medicines promoted the formation of multiocular smaller lipid droplets in adipocytes [82,83,85,86,87]. The activity of AMPK also contributes to the formation of smaller lipid droplets in adipocytes induced by olaparib [84]. In addition, overexpression of SIRT3 activated the AMPK pathway, which resulted in smaller lipid droplets size in 3T3-L1 adipocytes [89]. Consistent with these previous studies, our findings have also indicated that activation of AMPK can result in the formation of smaller lipid droplets in ADSCs. Therefore, targeting to activate AMPK could be considered a potential therapeutic candidate for obesity treatments. 

In summary, our results demonstrate that administration of metformin improves lipid profiles by decreasing the expression of Cidec, Perilipin1, and Rab8a involved in lipid droplets fusion and growth, followed by decreasing the activity and expression of lipogenic enzymes. We here also show that metformin decreases the expression of Cidec, Perilipin1, and Rab8a in part by activation of the AMPK pathway. Our study supports the development of clinical trials on metformin-based therapy for patients with overweight and obesity. In the meantime, the other possible contributing mechanisms of metformin are still being elucidated. Several recent studies have revealed that GDF15 mediates the effects of metformin on body weight and energy balance [22,23,24]. An exciting finding recently revealed that low-dose metformin binds PEN2 (a subunit of γ-secretase) and initiates an intersection signaling route to lysosomal AMPK activation, which ensures the therapeutic benefits of metformin without substantial side effects [90]. Further research of these pathways probably provides important insights into identifying new pharmacologic targets for obesity and other metabolic disorders. 

## 4. Materials and Methods

### 4.1. Animals

All animal experiments were performed in the animal facility of Capital Medical University (Beijing, China) and approved by the Animal Use and Care Committee of Capital Medical University. Sprague-Dawley male rats (180–200 g) were used in the study. The number of animals used was the minimum required for statistical analysis.

### 4.2. Materials

Collagenase type I was purchased from Worthington. Metformin was purchased from Enzo Life Sciences. 3-isobutyl-1-methylxanthine and dexamethasone were purchased from Sigma. Troglitazone was purchased from MCE. Insulin was purchased from Biotime Technology. Oleic acid was purchased from Santa crutz Biotechnology; Bodipy and protein assay kit were purchased from Thermos Fisher. RNA extraction kit was purchased from TIANgen Technologies, Beijing. Cell Counting Kit-8 kit, Triglyceride assay kit, RIPA, protease inhibitor cocktail, phosphatase inhibitor cocktail, ECL detection kit, and stripping buffer were purchased from Applygen Technologies, Beijing. 

### 4.3. Isolation and Differentiation of Adipose-Derived Stem Cells

Similar to previous descriptions [50,51,91,92], ADSCs were isolated from the epididymal and inguinal fat pads of male rats, respectively (denoted as Epi-ADSCs and Ing-ADSCs, respectively). The epididymal or inguinal fat pads were washed with Krebs-Ringer solution and minced in Krebs-Ringer solution containing 0.8 mg/mL collagenase type I and 1% free fatty acid-free bovine serum albumin. The fat pads were then digested on a horizontal shaking at 37 °C for 50 min. The digestive mixture was filtered through 80# and then 400# steel mesh. After centrifugation at 800× *g* for 10 min, cell pellets were collected, suspended, and counted. The isolated cells were maintained in Dulbecco’s Modified Eagle’s medium (DMEM)/F12 medium containing 10% FBS at 37 °C under an atmosphere of 5% CO_2_ and 95% air. For adipogenic differentiation, cells were incubated with differentiation medium I (DMI, DMEM/F12 supplemented with 5 µg/mL insulin, 5 mM 3-isobutyl-1-methylxanthine, 1 μM dexamethasone, 5 µM troglitazone and 10% FBS) for 2 days and differentiation medium II (DMII, DMEM/F12 supplemented with 5 µg/mL insulin, 5 µM troglitazone and 10% FBS) for another 2 days. After the induction of adipogenic differentiation, cells were maintained in DMEM/F12 containing 10% FBS for 4 days. The day of adding DMI was defined as the first day (day 0). Cells were treated with metformin at different concentrations during the whole adipogenic differentiation. The concentrations of metformin were determined according to previous studies [41,52,53,54]. In addition, cells were pretreated with AMPK inhibitor Dorsomorphin (Compound C) at 10 µM for 2 h, followed by incubation with metformin at 4 mM for the indicated time periods. 

### 4.4. Cell Viability Assay

Cell viability was determined by the Cell Counting Kit-8 (CCK-8) assay. Briefly, cells were plated at a density of 10^4^ cells/well in 96-well plates. Serial concentrations of metformin ranging from 1 mM to 8 mM were added into cell culture for 48 h. Cell viability was analyzed by using the CCK-8 kit according to the manufacturer’s protocol. 

### 4.5. Oil Red O and Bodipy Staining

After differentiation in the absence or presence of metformin at indicated times, cells were fixed with 4% paraformaldehyde and then washed with PBS. The cells were stained with 0.3% Oil Red O for 30 min. In addition, the accumulation of neutral lipids was examined using ImageJ software. If required, the stained Oil Red O in cells was extracted by absolute isopropanol, and the optical density was measured with a microplate reader at 490 nm. 

Bodipy staining was used to visualize and quantify intracellular lipid droplets. ADSCs were plated in confocal dishes and incubated with the differentiation medium. After treatments, adipocytes were incubated with 5 µM Bodipy staining solution containing 200 µM oleic acid for 16 h. The cells were then maintained in DMEM/F12 containing 10% FBS for another 1 h and examined using a confocal laser-scanning microscope. The diameter and number of lipid droplets in each adipocyte were analyzed with ImageJ software. 

### 4.6. Triglyceride Measurement

ADSCs were plated in 24-well plates and incubated with the differentiation medium with or without metformin. Then, the protein levels of intracellular triglyceride in adipocytes were assayed with a triglyceride assay kit according to the protocol (E1013, Applygen). The values were normalized to the corresponding protein concentration determined by BCA protein assay kit (Thermo Scientific, Rockford, IL, USA).

### 4.7. Quantitative Real-Time PCR

Total RNA was extracted from the cells at indicated times with a RNA extraction kit (TIANgen, China) according to the manufacturer’s instructions. qRT-PCR was performed with SuperReal PreMix Plus (SYBR Green) (TIGANGEN, Beijing, China) on CFX96 Real-time System, C1000 Thermal Cycler (BioRad). The primers used are available in Table 1. The values were determined by comparison with the expression of β-actin RNA as an internal control. 

### 4.8. Immunofluorescent Staining

ADSCs were fixed for immunofluorescent staining as in previous descriptions [93]. The primary antibody was guinea pig anti-Perilipin1 (GP29, 1:250, Progen). The secondary antibody was Alexa Fluor-594-conjugated goat anti-guinea pig IgG (106-295-003, 1:200, Jackson Immunoresearch Laboratories, Inc., West Grove, PA, USA). The cells were examined using a confocal laser-scanning microscope. 

### 4.9. Immunoblotting

ADSCs were lysed and sonicated in RIPA buffer containing protease inhibitor cocktail and phosphatase inhibitor cocktail on ice. After centrifugation, the supernatant was collected, and the protein concentration was determined by the BCA protein assay kit (Thermo Scientific). After SDS-PAGE electrophoresis, proteins were transferred to PVDF membranes. After blocking with 5% milk or 5% BSA at room temperature for 1 h, incubation with the following primary antibodies was performed overnight at 4 °C. The primary antibodies were anti-β-actin (ab8227, 1:1000, Abcam, Cambridge, UK), anti-p-AMPKα (CST2535, 1:1000, CST, Danvers, Massachusetts, USA), anti-AMPKα (CST5831, 1:1000, CST), anti-Cidec (ab213693, 1:500, Abcam), anti-Rab8a (610844, 1:1000, BD), anti-Perilipin1 (GP29, 1:1000, Progen, Heidelberg, Germany), anti-Pparγ (CST2443, 1:1000, CST). All membranes were then incubated with the corresponding HRP-conjugated secondary antibodies and developed using ECL reagents. β-actin was used as the loading control. If required, a stripping buffer was used to remove the primary antibodies from the membranes. After stripping, membranes were blocked and reprobed with other primary antibodies. The band intensities were analyzed by the Image J software (NIH, Bethesda, MD, USA). β-actin was used as an internal control.

### 4.10. Statistical Analysis

All experiments were performed in triplicate. All Data are expressed as the mean ± SEM. All statistical analyses were performed using GraphPad Prism 6.0. The data were analyzed by one-factor or two-factor ANOVA followed by Tukey’s test. Statistical difference was showed in each figure as * *p* < 0.05, ** *p* < 0.01, and *** *p* < 0.001.

## Figures and Tables

**Figure 1 ijms-23-05986-f001:**
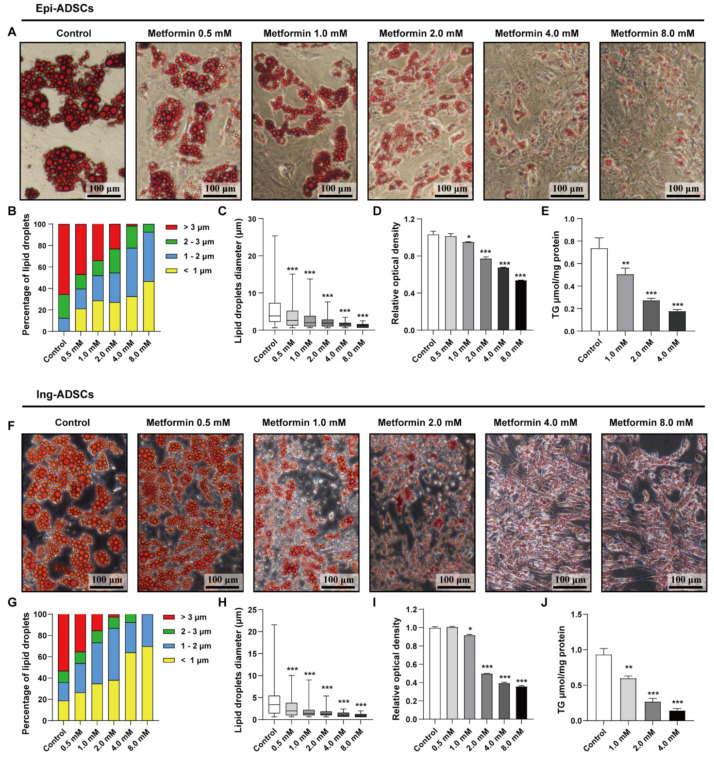
Lipid accumulation in the process of ADSCs differentiation after metformin treatments. (**A**) The images showing Oil Red O staining of differentiated adipocytes from Epi-ADSCs cultures with serial doses of metformin. (**B**–**E**) The diagrams showing the percentage of lipid droplets diameter (**B**), average lipid droplets diameter (**C**), relative optical density of isopropanol extraction (**D**), and triglyceride deposition (**E**) of differentiated adipocytes from Epi-ADSCs cultures with serial doses of metformin. (**F**) The images showing Oil Red O staining of differentiated adipocytes from Ing-ADSCs cultures with serial doses of metformin. (**G**–**J**) The diagrams showing the percentage of lipid droplets diameter (**G**), average lipid droplets diameter (**H**), relative optical density of isopropanol extraction (**I**), and triglyceride deposition (**J**) of differentiated adipocytes from Ing -ADSCs cultures with serial doses of metformin. Scale bars represent 100 μm for A and F. A one-factor ANOVA followed by a Tukey’s test was used to make group comparisons. * *p* < 0.05, ** *p* < 0.01, *** *p* < 0.001 versus control group. Epi-ADSCs, adipose-derived stem cells (ADSCs) from the epididymal fat pads; Ing-ADSCs, adipose-derived stem cells (ADSCs) from the inguinal fat pads; TG, triglyceride.

**Figure 2 ijms-23-05986-f002:**
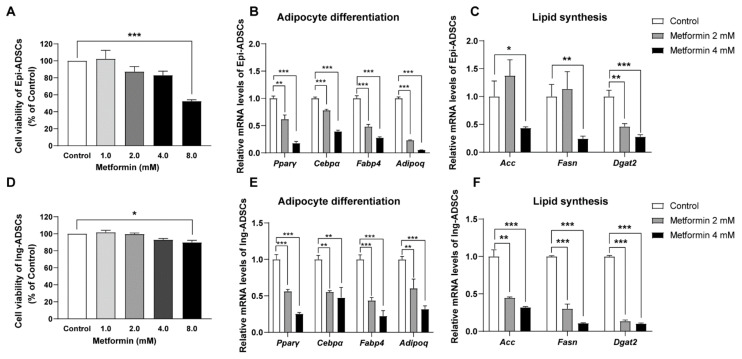
Downregulation of adipogenic genes and lipogenic enzymes in ADSCs differentiation after metformin treatments. (**A**) CCK8 assay showing cell viability in Epi-ADSCs with metformin treatments. (**B**,**C**) Quantitative real-time PCR showing relative mRNA levels of adipogenic transcription factors (**B**) and lipogenic enzymes (**C**) in differentiated Epi-ADSCs with metformin treatments. (**D**) CCK8 assay showing cell viability in Ing-ADSCs with metformin treatments. (**E**,**F**) Quantitative real-time PCR showing relative mRNA levels of adipogenic transcription factors (E) and lipogenic enzymes (**F**) in differentiated Ing-ADSCs with metformin treatments. A one-factor ANOVA followed by a Tukey’s test was used to make group comparisons. * *p* < 0.05, ** *p* < 0.01, *** *p* < 0.001 versus control group. Acc, acetyl-CoA carboxylase; C/EBPα, CCAAT/enhancer-binding protein-α; Dgat2, diacylglycerol acyltransferase 2; Epi-ADSCs, adipose-derived stem cells (ADSCs) from the epididymal fat pads; Fabp4 fatty acid-binding protein 4; Fasn, fatty acid synthesis. Ing-ADSCs, adipose-derived stem cells (ADSCs) from the inguinal fat pads; LD, lipid droplets; Pparγ, peroxisome proliferator-activated receptor γ.

**Figure 3 ijms-23-05986-f003:**
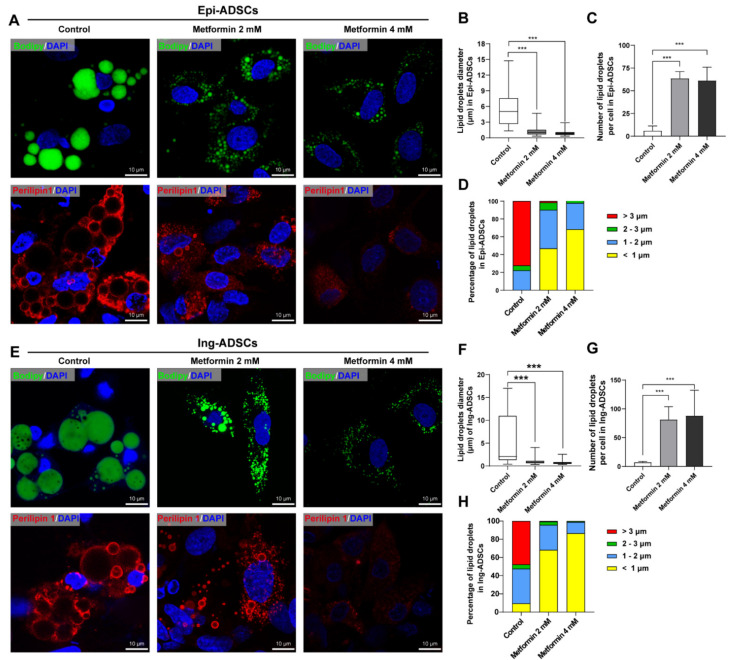
The characteristics of lipid droplets of differentiated adipocytes after metformin treatments. (**A**) Representative images showing characteristics of lipid droplets (Bodipy staining, green, upper panel; Immunofluorescence of Perilipin1 staining, red, lower panel; DAPI, blue) in differentiated adipocytes from Epi-ADSCs cultures after metformin treatments. (**B**–**D**) The diagrams showing the lipid droplets diameter per cell (**B**), the number of lipid droplets per cell (**C**), and the percentage of lipid droplets (**D**) in differentiated Epi-ADSCs with metformin treatments. (**E**) Representative images showing characteristics of lipid droplets (Bodipy staining, green, upper panel; immunofluorescence of Perilipin1 staining, red, lower panel; DAPI, blue) in differentiated adipocytes from Ing-ADSCs cultures after metformin treatments. (**E**–**H**) The diagrams showing the lipid droplets diameter per cell (**F**), the number of lipid droplets per cell (**G**), and the percentage of lipid droplets (**H**) in differentiated Ing-ADSCs with metformin treatments. Scale bars represent 10 μm for A and E. A one-factor ANOVA followed by a Tukey’s test was used to make group comparisons. *** *p* < 0.001 versus control group. Epi-ADSCs, adipose-derived stem cells (ADSCs) from the epididymal fat pads; Ing-ADSCs, adipose-derived stem cells (ADSCs) from the inguinal fat pads.

**Figure 4 ijms-23-05986-f004:**
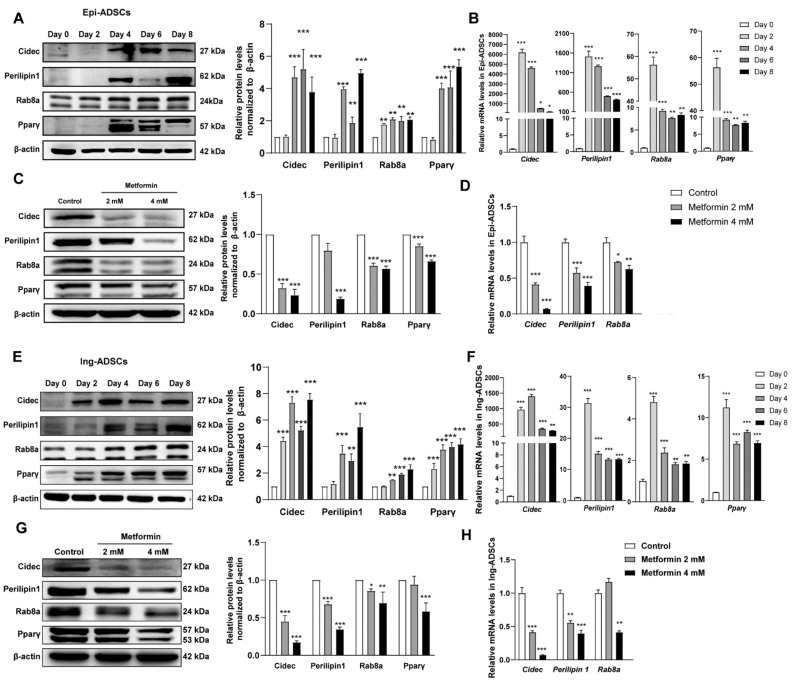
Expression of Cidec and its regulatory factors in ADSCs differentiation after metformin treatments. (**A**–**D**) Relative Cidec, Perilipin1, Rab8a, and Pparγ protein levels and mRNA levels in the differentiation process of Epi-ADSCs cultures (**A**,**B**) and differentiated adipocytes from Epi-ADSCs cultures after metformin treatments (**C**,**D**). (**E**–**H**) Relative Cidec, Perilipin1, Rab8a, and Pparγ protein levels and mRNA levels in the differentiation process of Ing-ADSCs cultures (**E**,**F**) and differentiated adipocytes from Ing-ADSCs cultures after metformin treatments (**G**,**H**). A one-factor ANOVA followed by a Tukey’s test was used to make group comparisons. * *p* < 0.05, ** *p* < 0.01, *** *p* < 0.001 versus control group or group at day 0. Cidec, cell death-inducing DFFA-like effector C; Epi-ADSCs, adipose-derived stem cells (ADSCs) from the epididymal fat pads; Ing-ADSCs, adipose-derived stem cells (ADSCs) from the inguinal fat pads; Pparγ, peroxisome proliferator-activated receptor γ; Rab8a, ras-related protein 8a.

**Figure 5 ijms-23-05986-f005:**
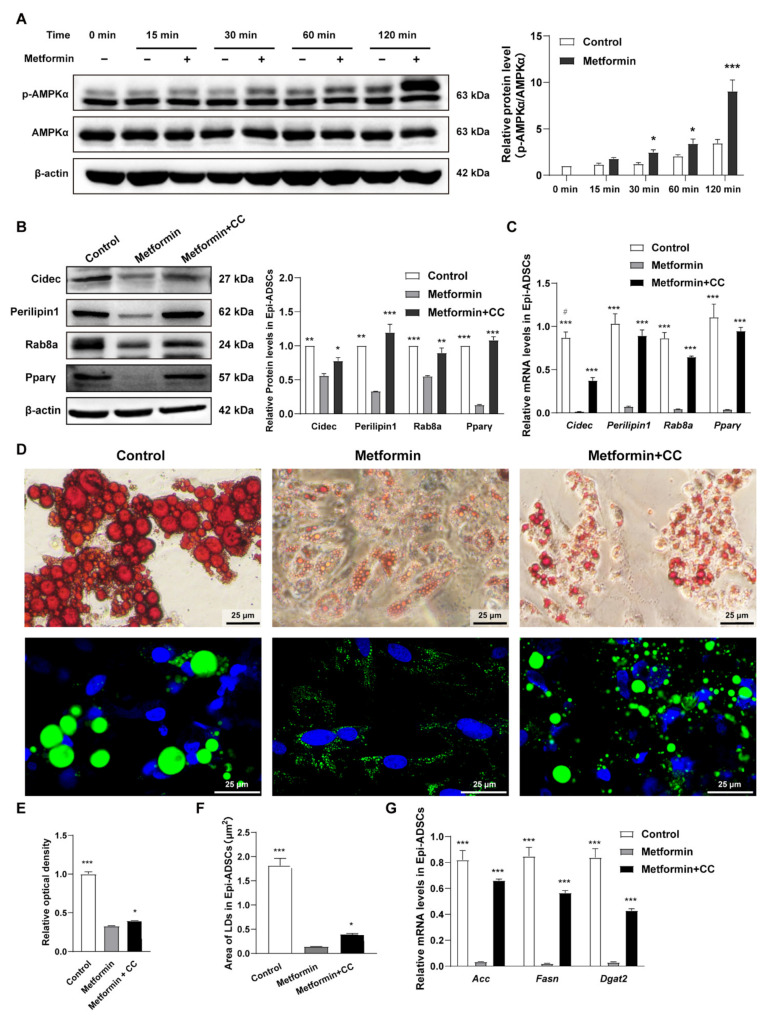
Metformin inhibits Cidec and its regulatory factors expression partly via AMPK activation in Epi-ADSCs cultures. (**A**) Western blot showing relative p-AMPKα protein levels in Epi-ADSCs cultures after metformin treatments at the indicated time points. (**B**) Western blot showing relative protein levels and (**C**) Quantitative real-time PCR showing relative mRNA levels of Cidec, Perilipin1, Rab8a, and Pparγ in Epi-ADSCs cultures after metformin treatments with or without CC. (**D**) Representative images showing the morphology of lipid droplets (Oil Red O staining, upper panel; Bodipy staining, green, DAPI, blue, lower panel) in Epi-ADSCs cultures after metformin treatments with or without CC. Scale bars represent 25 μm for D. (**E**–**G**) The diagrams showing the relative optical density of isopropanol extraction (**E**), area of LDs (**F**), and relative mRNA levels of Acc, Fasn, and Dgat2 expression in differentiated adipocytes from Epi-ADSCs cultures after metformin treatments with or without CC. A one-factor or two-factor ANOVA followed by a Tukey’s test was used to make group comparisons. * *p* < 0.05, ** *p* < 0.01, *** *p* < 0.001 versus metformin group or group at corresponding indicated time points. ^#^
*p* < 0.01 versus metformin + CC group. Acc, acetyl-CoA carboxylase; CC, Compound C; Cidec, cell death-inducing DFFA-like effector C; Dgat2, diacylglycerol acyltransferase 2; Epi-ADSCs, adipose-derived stem cells (ADSCs) from the epididymal fat pads; Fasn, fatty acid synthesis; LDs, lipid droplets; Pparγ, peroxisome proliferator-activated receptor γ; Rab8a, ras-related protein 8a.

**Figure 6 ijms-23-05986-f006:**
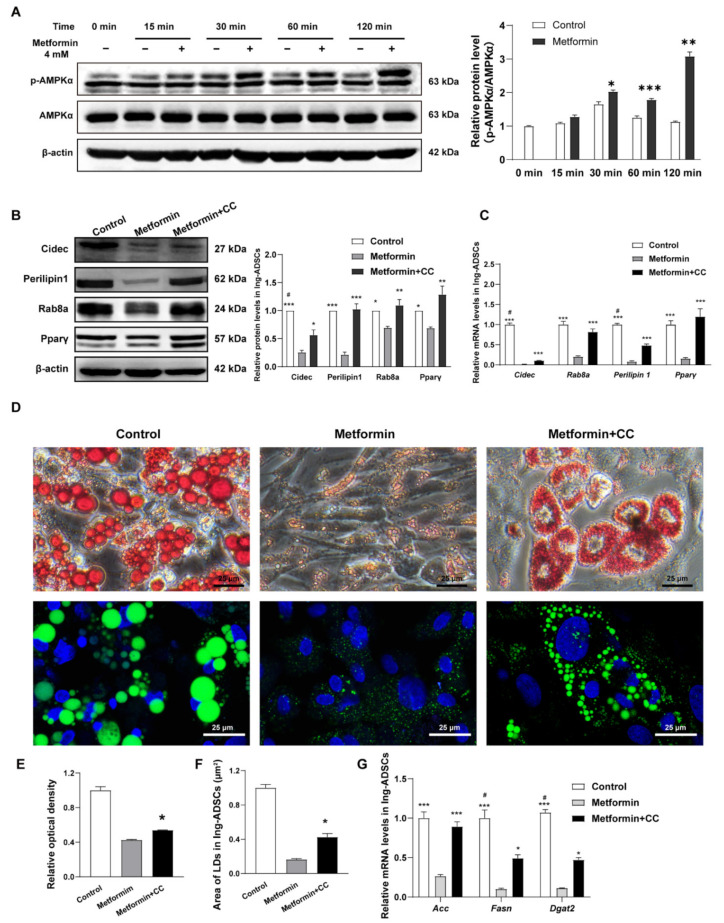
AMPK activation partly involves Metformin inhibition on Cidec and its regulatory factors expression in Ing-ADSCs cultures. (**A**) Western blot showing relative p-AMPKα protein levels in Ing -ADSCs cultures after metformin treatments at the indicated time points. (**B**) Western blot showing relative protein levels and (**C**) Quantitative real-time PCR showing relative mRNA levels of Cidec, Perilipin1, Rab8a, and Pparγ in Ing-ADSCs cultures after metformin treatments with or without CC. (**D**) Representative images showing the morphology of lipid droplets (Oil Red O staining, upper panel; Bodipy staining, green, DAPI, blue, lower panel) in Ing -ADSCs cultures after metformin treatments with or without CC. Scale bars represent 25 μm for D. (**E**,**F**) The diagrams showing the relative optical density of isopropanol extraction (**E**), average area of lipid droplets (**F**) and relative mRNA levels of Acc, Fasn, and (**G**) Dgat2 expression in differentiated adipocytes from Ing-ADSCs cultures after metformin treatments with or without CC. A one-factor or two-factor ANOVA followed by a Tukey’s test was used to make group comparisons. * *p* < 0.05, ** *p* < 0.01, *** *p* < 0.001 versus metformin group or group at corresponding indicated time points. ^#^
*p* < 0.01 versus metformin + CC group. Acc, acetyl-CoA carboxylase; CC, Compound C; Cidec, cell death-inducing DFFA-like effector C; Dgat2, diacylglycerol acyltransferase 2; Ing-ADSCs, adipose-derived stem cells (ADSCs) from the inguinal fat pads; Fasn, fatty acid synthesis; LDs, lipid droplets; Pparγ, peroxisome proliferator-activated receptor γ; Rab8a, ras-related protein 8a.

**Table 1 ijms-23-05986-t001:** Primers used for quantitative real-time PCR.

Genes	Forward Sequence (5′-3′)	Reverse Sequence (5′-3′)
Pparγ	TCTTAACTGTCGGATCCAC	CAAACCTGATGGCATTGTGA
Cebpa	AGATAAAGCCAAACAGCGCAAC	CCTAGAGATCCAGCGACCCT
Fabp4	CCGAGATTTCCTTCAAACTGG	ATGCCCTTTCGTAAACTCT
Adipoq	GTCACTGTCCCCAATGTTCC	TTTTCCTGATACTGGTCGT
Acc	ACTTTAACCGTGAAGGGCTA	CCATCCACAATATAAGCACCA
Fasn	CCATTTCCATTGCCCTTAGCC	GTAACACATGCTGCTCAAACGA
Dgat2	ATAGCTGCTCTCTACTTCACC	AGTCTCGAAAATAGCGCCACA
Cidec	AGCCCTACCAAGAGATACAGT	CGTCGATCTTCTTAGTTGGCTT
Perilipin1	CCATGTCCCTATCCGATGCC	ATGTCTCGGTTTTGTCATCCAG
Rab8a	CTCTAACCCTTCCCCGATG	ATGACTATTGCTGGTACGTT
β-actin	ATCCGTAAAGACCTCTATGCC	GCTCAGTAACAGTCCGCCTA

## Data Availability

Not applicable.

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
