# Peer review of "Metformin Inhibits Lipid Droplets Fusion and Growth via Reduction in Cidec and Its Regulatory Factors in Rat Adipose-Derived Stem Cells"

_ijms, 2022, doi:10.3390/ijms23115986_

Round 1

Reviewer 1 Report

Dear Authors,

After carrying out and a detailed reading of the scientific article prepared by you, I was able to perceive through the experimental protocols used, the investigation techniques, the self-citations carried out in a completely justified way, since I read the other articles produced by the research group (DOI: 10.1152/ajpendo.00049.2017; doi: 10.1016/j.molmet.2021.101400; doi: 10.1016/j.bbalip.2020.158871) in which you are inserted and I noticed that the group is already dedicated to the study of topics related to the content addressed and discussed in this article.
I understand that the main results obtained and described by the researchers responsible for preparing this article:
1) Metformin reduces lipid accumulation in both Ing-ADSCs and Epi-ADSCs cultures;
2) Metformin decreases adipogenic genes and lipogenic enzymes expression in ADSCs differentiation;
3) Metformin inhibits lipid droplets fusion and growth in ADSCs differentiation;
4) Cidec, perilipin 1 and Rab8a expression are decreased in ADSCs after metformin treatments;
5) Metformin inhibits Cidec, perilipin 1 and Rab8a expression partly via activation of AMPK signaling in ADSCs
These results are very interesting, enlightening and consistently answer the questions related to the objectives proposed by the authors and, in addition, the data described in the article were obtained through the use of adequate experimental techniques. In addition, these results allow a better understanding of the reasons why metformin promotes weight loss in many patients undergoing pharmacotherapy with this drug.
However, I would like to know why the text of lines 99 to 109 is written in bold, is there any special reason for this?
I conclude the review of the article by congratulating you for the excellent work you have done and, sincerely, I hope that this work will inspire the realization of clinical studies and new research that result in the discovery of new molecular targets for the action of drugs that aim to promote the treatment of obese patients. is of fundamental importance, considering that obesity, nowadays, has become a serious public health problem, which causes the loss of many lives and entails a high economic cost to the authorities responsible for the administration of the health systems of rich and poor countries. Worldwide.
Thank you very much for the honorable invitation and attention.

Kind regards,

Reviewer 2 Report

This manuscript contains extensive amount of in vitro data which support the inhibitory effect of metformin on the accumulation of lipid droplets in differentiated mouse adipocytes of two distinct anatomical origins. The Authors also provide mechanistic data to explain the described observations. The data presented here is clinically relevant and the manuscript is well written. However, a few points should be clarified before the final publication of the manuscript.

Specific suggestions:

  1. It should be included in the title that the observed phenomenon was found in mice.
  2. Due to the current preference in scientific literature of a non-stigmatizing language to describe diseases, adjectives should be avoided. Therefore, instead of “obese patients”, “patients with obesity” should be written.
  3. L194: “Fig. 2E-F” instead of “Fig. 2D-F”.
  4. The PPARg mRNA expression data was displayed twice (Figures 2B/E and 4D/H).
  5. L279-281: The statement requires a reference.
  6. L285-287: I could not find the data described in this sentence.
  7. Several studies indicated that the activation of AMPK induces beige adipogenesis which results in the formation of smaller lipid droplets. This should be discussed in the manuscript.
  8. L402-403: The sentence should be rephrased for clarification.
  9. 4.8. and 4.9. Catalogue numbers and/or working dilutions of the antibodies should be indicated.
  10. All abbreviations should be given when first mentioned in the text.

Round 2

Reviewer 2 Report

The manuscript was substantially improved. Therefore, I suggest the current version of this manuscript for publication in IJMS.